# Promising Effect of a New Ketogenic Diet Regimen in Patients with Advanced Cancer

**DOI:** 10.3390/nu12051473

**Published:** 2020-05-19

**Authors:** Keisuke Hagihara, Katsufumi Kajimoto, Satoshi Osaga, Naoko Nagai, Eku Shimosegawa, Hideyuki Nakata, Hitomi Saito, Mai Nakano, Mariko Takeuchi, Hideaki Kanki, Kuriko Kagitani-Shimono, Takashi Kijima

**Affiliations:** 1Department of Advanced Hybrid Medicine, Osaka University Graduate School of Medicine, Osaka 565-0871, Japan; kkajimot@gmail.com (K.K.); nakata.hideyuki@gmail.com (H.N.); hitomis.og@gmail.com (H.S.); norse.vsta.m@gmail.com (M.N.); takemari@neurol.med.osaka-u.ac.jp (M.T.); kanki@neurol.med.osaka-u.ac.jp (H.K.); 2Clinical Research Management Center, Nagoya City University Hospital, Aichi 467-8602, Japan; crohsaga@med.nagoya-cu.ac.jp; 3Division of Nutritional Management, Osaka University Hospital, Osaka 565-0871, Japan; nagaink@hosp.med.osaka-u.ac.jp; 4Department of Molecular Imaging in Medicine, Osaka University Graduate School of Medicine, Osaka 565-0871, Japan; eku@mi.med.osaka-u.ac.jp; 5United Graduate School of Child Development, Osaka University, Osaka 565-0871, Japan; kuriko@ped.med.osaka-u.ac.jp; 6Department of Respiratory Medicine and Hematology, Hyogo College of Medicine, Hyogo 663-8501, Japan; tkijima@hyo-med.ac.jp

**Keywords:** cancer, ketogenic diet, PET-CT

## Abstract

A ketogenic diet is expected to be an effective support therapy for patients with cancer, but the degree and duration of carbohydrate restriction are unclear. We performed a case series study of a new ketogenic diet regimen in patients with different types of stage IV cancer. Carbohydrates were restricted to 10 g/day during week one, 20 g/day from week two for three months, and 30 g/day thereafter. A total of 55 patients participated in the study, and data from 37 patients administered the ketogenic diet for three months were analyzed. No severe adverse events associated with the diet were observed. Total ketone bodies increased significantly, and both fasting blood sugar and insulin levels were suppressed significantly for three months after completion of the study. Five patients showed a partial response on Positron emission tomography–computed tomography (PET-CT) at three months. Three and seven patients showed complete and partial responses, respectively at one year. Median survival was 32.2 (maximum: 80.1) months, and the three-year survival rate was 44.5%. After three months on the ketogenic diet, the serum Alb, BS, and CRP (ABC) score could be used to stratify the patients into groups with significantly different survival rates (*p* < 0.001, log-rank test). Our ketogenic diet regimen is considered to be a promising support therapy for patients with different types of advanced cancer.

## 1. Introduction

Cancer is one of the most important issues in medicine. Cancer treatments, such as immune checkpoint inhibitors and molecular target drugs, have developed in this decade, and the prognosis of cancer patients has improved [1]. At the same time, cancer patients have had to take various cancer therapies with risks of malnutrition and metabolic derangements [2]. Weight loss in gastrointestinal cancer patients with chemotherapy has been found to be associated with overall survival [3]. Obesity and performance status were also found to be independently associated with malnutrition, which was related to mortality of cancer patients [4]. Increased serum levels of insulin and especially glucose were found to increase the risk of colorectal adenoma recurrence [5]. Nutritional therapies have been expected to improve the quality of life and prognosis of cancer patients [2].

The ketogenic diet, a high fat, low carbohydrate, and adequate protein regimen, has been used as second-line treatment for children with seizure syndrome [6]; it reduces phosphatidylinositol 3-kinase (PI3K) signaling by lowering fasting insulin levels in the brain [7]. PI3K plays an important role in cancer growth, and several genomic alterations in PI3K signaling have been identified [7]. The Warburg effect is well known in cancer cells, which predominantly use glucose if oxygen is present [8]. Metabolic alterations are widespread in cancer cells [9], and the ketogenic diet is a promising nutritional intervention to improve them. Several preclinical studies in cancer model mice showed a combined effect of the ketogenic diet and radiation or chemotherapy [10,11]. The effect of the ketogenic diet with conventional therapy was reported in cases of glioblastoma [12,13]. The ketogenic diet was also shown to improve quality of life in several types of advanced cancers [14]. However, the results of early clinical trials of the ketogenic diet are controversial [10,11]. Several problems still remain in the use of the ketogenic diet in cancer patients, because it has been used primarily for children with epilepsy. The traditional ketogenic diet has a ketone ratio of 4:1. The patients take 90% of their calories from fat, and most studies showed poor adherence [10]. The appropriate carbohydrate restriction, ketone ratio, and duration of the ketogenic diet are unclear; the ketogenic diet regimen should be optimized for adult cancer patients. In addition, factors that can predict clinical responses with the ketogenic diet are unclear. Thus, a new ketogenic diet regimen showing stable adherence and a clinical effect is presented, along with factors that can predict its effects patients with cancer.

## 2. Materials and Methods

### 2.1. Design and Participants

A case series study to investigate the efficacy and safety of a new ketogenic diet regimen in advanced cancer patients was conducted between February 2013 and December 2018 (follow-up November 2019). This study was approved by the genome committee of Osaka University (Approval number 526). Participants attending the regional core hospital for cancer therapy were referred to our department. Entry criteria were as follows: patients diagnosed with cancer by histology or cytology and evaluated as stage IV on computed tomography (CT), magnetic resonance imaging (MRI), or surgery by their attending oncologist; performance status <2; and could take foods orally. The exclusion criteria were as follows: could not ingest foods orally; performance status >3; and diabetes mellitus. The study complied with the Declaration of Helsinki II. Informed consent was obtained from all participants

### 2.2. Ketogenic Diet for Cancer Patients

We discussed a ketogenic diet regimen for cancer patients with a pediatrician and registered dieticians who have treated children with refractory epilepsy using a ketogenic diet in Osaka University Hospital [15]. After consideration of safety and feasibility, a ketogenic diet regimen was developed for cancer patients (Figure 1A). Carbohydrate was restricted to 10 g/day during week 1. The amounts of lipid and protein were adjusted according to the ketone ratio (lipid/protein and carbohydrates) target of 2:1. The calorie intake was set to 30 kcal/kg/day based on the patients’ real weight. Assuming that the ideal weight of the patient is 50 kg (the same as below), total calorie intake is 1500 kcal, carbohydrate intake is 10 g, lipid intake is 140 g, and protein intake is 60 g (ketone ratio 2:1). A representative menu of the ketogenic diet for one day is shown in Appendix A. From week two, for three months, carbohydrate was restricted to 20 g/day. Total calorie intake was 1400–1600 kcal/day, lipid intake was 120–140 g, and protein intake was 70 g (ketone ratio 1:1). Three months later, for patients who wished to continue the ketogenic diet, restriction of carbohydrate was 30 g/day or less (≤10 g/one meal). Required trace elements and vitamins were provided by supplements. For calorie supplementation, medium chain triglyceride (MCT) oil (about 50–80 g/day) and a ketogenic formula (about 30 g/day) were used. The MCT oil and ketogenic formula (Ketonformula^®^ 817-B) were provided as gifts from Nisshin Oillio (Tokyo, Japan) and Meiji Co. Ltd. (Tokyo, Japan), respectively. The contents of a ketogenic formula were shown in Appendix A. The patients began the ketogenic diet under the supervision of registered dieticians from the division of Nutritional Management, Osaka University Hospital, after obtaining the patients’ informed consent. Registered dieticians provided nutritional guidance according to general condition and laboratory data at enrollment, at 7 days after enrollment, and at months 1, 2, and 3. After that, patients received nutritional guidance every 1–2 months. Combinations with chemotherapy, radiation therapy, and surgery were possible as required.

### 2.3. Evaluation and Statistical Analysis

Tumor size was evaluated by positron emission tomography–computed tomography(PET-CT) before and three months after starting the ketogenic diet. The clinical response was evaluated one year after starting the ketogenic diet, and overall survival was determined. To examine the safety of the ketogenic diet for cancer patients, the QOL score (EORTC QLQ-C30), gastrointestinal symptoms score (GSRS), blood tests, and assessments of body composition using In body 720 (Biospace Co. Ltd., Seoul, Korea) were performed. The patients’ complete blood cell counts, biochemistry, urinalysis, and blood ketone bodies were evaluated every month. All blood data and assessments of body composition were analyzed by the Wilcoxon signed-rank test. All questionnaire results were analyzed by paired *t*-tests. The relationships between blood data and prognosis were examined by log-rank tests. Statistical analysis was performed by an independent biostatistician. All adverse events according to Common Terminology Criteria for Adverse Events (CTCAE) v5.0 were counted from enrollment to month 3.

### 2.4. PET-CT

All patients underwent 18F-FDG PET before and 12–16 weeks after starting the ketogenic diet for cancer patients in our hospital. PET/CT was performed with an integrated scanner (Gemini GXL; Philips, Amsterdam, The Netherlands). Whole-body images, generally from the top of the skull to the mid-thigh level, were acquired about 60 min after intravenous injection of 18F-FDG at a dose of 3.7 MBq (0.10 mCi)/kg body weight. The therapeutic response on a ketogenic diet for cancer patients was evaluated using the response evaluation criteria in solid tumors (RECIST) version 1.1 criteria based on CT measurements [16] and the European Organization for Research and Treatment of Cancer (EORTC) criteria based on PET/CT measurements [17]. All images were analyzed by two experienced nuclear medicine physicians. 

## 3. Results

Fifty-five patients gave their informed consent between February 2013 and November 2018, and their background characteristics are shown in Appendix A. Finally, data from 37 patients who took the ketogenic diet for three months were analyzed. Five patients did not take the ketogenic diet. Eleven patients dropped out for various reasons (Figure 1B). Two patients were excluded due to difficulty evaluating the PET-CT findings (Figure 1B). The 37 patients included eight colorectal cancer patients, six lung cancer patients, five breast cancer patients, four pancreatic cancer patients, and 19 other cancer patients (Table 1). Thirty-three patients (89.2%) received chemo-hormonal therapy. More detailed histological and therapeutic background characteristics of the 37 patients are shown in Appendix A.

After starting the ketogenic diet, acetoacetic acid and β-hydroxybutyric acid (BHB) levels were significantly increased from one week to three months (139.9 ± 178.0 to 922.1 ± 340.1, 573.2 ± 356.0 μmol/L, *p* < 0.001, 333.1 ± 178.0 to 2678.5 ± 1195.9, 1697.8 ± 1287.8 μmol/L, *p* < 0.001, respectively, Wilcoxon signed-rank test). Both fasting blood sugar and insulin levels were significantly suppressed for three months after completing the study (96.8 ± 11.5 to 91.2 ± 13.0 mg/dL, *p* < 0.01, 8.6 ± 11.4 to 4.8 ± 4.6 μIU/mL, *p* < 0.05, respectively, Wilcoxon signed-rank test). Serum albumin (Alb) and C-reactive protein (CRP) levels showed no significant changes (Figure 2).

The glucose ketone index (GKI) has been proposed as an index to judge the state of ketosis [18]. The GKI is calculated as the molar ratio of glucose and BHB. The GKI is separated into five levels of ketosis (Figure 3A). A more than moderate GKI is said to be functional ketosis. In the present study, the GKI showed moderate to high ketosis in 70% of the patients for three months (Figure 3B). We showed glucose, BHB, and GKI values for each of 37 patients in detail (Appendix A).

Tumor size was evaluated at three months using PET-CT. Representative data of the PET-CT findings that showed markedly decreased 18F-FDG hotspots after the ketogenic diet are presented. A 58-year-old male colorectal cancer patient with T3N0M1a, Stage IV, tubular adenocarcinoma underwent sigmoidectomy for ileus and was then treated with mFOLFOX6 and Pmab for multiple liver metastases. Three months after starting the ketogenic diet with chemotherapy, the multiple liver metastases were dramatically decreased (Figure 4A). A 55-year-old male lung cancer patient with cT3N1M1a, Stage IVA, EGFR (−), ROS1 (+) adenocarcinoma presented with a pleural effusion and was treated with crizotinib 250 mg twice daily and referred to our department. Four months after starting the ketogenic diet, the left 18F-FDG hotspots were dramatically diminished (Figure 4B). Five patients showed a partial response on PET-CT at three months. Examining the clinical response one year after starting the ketogenic diet, three and seven patients had complete and partial responses, respectively. The response rate was 27.0% (Figure 4C). Moreover, the ketogenic diet markedly improved the patients’ long-term prognoses. The median overall survival was 32.2 (maximum 80.1) months, and the 3-year survival rate was 44.5% (Figure 5A). The median survival in more than 4 participants was 41.1 months in breast cancer patients, 19.0 months in colorectal cancer patients, and 10.7 months in pancreatic cancer patients. In non-small-cell lung cancer patients, the median survival time was not reached (median observation time 26.8 months).

The relationships between clinical data and prognosis were examined. Alb and CRP are well known as predictors of prognosis in cancer patients [19]. In the present study, Alb and CRP were predictors in cancer patients, as in previous reports (Figure 5B,D). The relationships among acetoacetic acid, BHB, insulin, BS, and prognosis were also examined. Surprisingly, BS could stratify the prognosis of cancer patients treated with the ketogenic diet (Figure 5C). Other factors did not show significant differences (data not shown). The ketogenic diet-ABC (KD-ABC) score, consisting of Alb, BS, and CRP (ABC) values, was then examined after three months on the ketogenic diet to determine whether it could predict the therapeutic response (Figure 5E). It was found that the KD-ABC scores could stratify the patients into groups with significantly different survival rates (*p* < 0.001, log-rank test, Figure 5F).

The safety of the present ketogenic diet for cancer patients was evaluated. Total GSRS showed no significant difference between before and three months after treatment (1.64 ± 0.53 to 1.80 ± 0.76). The constipation score was significantly increased at one and two months and recovered to baseline at three months. The reflux score increased significantly (1.23 ± 0.40 to 1.60 ± 0.90, *p* < 0.05, paired *t*-test) (Appendix A). On the EORTC-QLQ—c30, the global health score showed no significant difference between before and three months after treatment (59.7 ± 24.7 to 56.5 ± 26.9). The physical conditioning and dyspnea scores were both significantly changed at three months (86.0 ± 15.6 to 79.4 ± 20.7, 11.8 ± 16.2 to 20.4 ± 20.5, respectively; both *p* < 0.05, paired *t*-test). The constipation score showed the same results as the GSRS (Appendix A).

The adverse events of 55 patients from the start to three months after the ketogenic diet were evaluated. A total of 1143 adverse events were analyzed; 868 events were due to worsening of cancer or chemotherapy, and 275 adverse events were related to the ketogenic diet and are summarized in Table 2. None of the patients developed severe adverse events associated with the ketogenic diet. There were no cases of leukocytopenia, anemia, or thrombocytopenia. Similarly, no deteriorations in liver and kidney functions were observed (Appendix A). On the other hand, hyperuricemia and hyperlipidemia occurred in almost half of the 55 patients. Although serum uric acid was significantly increased from one week to three months (5.1 ± 1.2 to 8.6 ± 2.4, 6.0 ± 1.8 mg/dL, *p* < 0.001, Wilcoxon signed-rank test), no patients developed attacks of gout. The hyperuricemia was normalized with drugs or observation. Total cholesterol and low-density lipoprotein cholesterol levels were significantly increased from one week to three months (211.5 ± 57.9 to 268.8 ± 76.2 mg/dL, *p* < 0.001, 126.4 ± 47.9 to 161.5 ± 66.3 mg/dL, *p* < 0.001, respectively, Wilcoxon signed-rank test). However, no cardiovascular events occurred, and hyperlipidemia was well controlled by anti-hyperlipidemic drugs. Grade 2 levels of constipation, stomach pain, and diarrhea were reported, but they were easily relieved using several drugs. Weight loss developed in 12 patients. However, the ketogenic diet significantly decreased fat mass (12.4 ± 6.4 to10.3 ± 4.8 kg, *p* < 0.001, Wilcoxon signed-rank test), but not muscle weight (17.4 ± 4.8 to 17.2 ± 4.7 kg, *p* = 0.169, Wilcoxon signed-rank test) (Appendix A). Taken together, the present ketogenic diet regimen for cancer patients appeared to be tolerable and safe in the treatment of cancer. 

## 4. Discussion

The ketogenic diet is expected to be effective for cancer. In fact, the possibility of using the ketogenic diet was examined in various types of cancer model mice [10,11]. The effects of the ketogenic diet alone differ depending on the type of cancer, but the combined effect of the ketogenic diet with chemotherapy or radiation is promising [10,11]. On the other hand, the results of clinical trials of a ketogenic diet for cancer patients were controversial [10,11]. This is because strict dietary restrictions, e.g., a ketone ratio of 2:1 to 4:1, cannot be continued for adult cancer patients. If the carbohydrate restriction is insufficient, for example glucose restriction to 50–70 g/day, serum ketone bodies are not fully induced, and the anti-tumor effect is unclear. Moreover, it is not clear how long patients have to continue the ketogenic diet to show the anti-tumor effect.

The present study demonstrated that the ketogenic diet regimen showed stable adherence and induced functional ketosis with high reproducibility and was well controlled in advanced cancer patients receiving chemotherapy. The clinical effect of the ketogenic diet in cancer patients was evaluated using PET-CT and long-term observation, and it was found that the present ketogenic diet regimen seems to have an effect on long-term survival of advanced cancer patients compared with the previous reports [10,11].

Although various cancer treatments have been developed, the 5-year relative survival rates of lung cancer and pancreatic cancer patients with distant metastases were 5% and 3%, respectively [1]. The median overall survival with erlotinib treatment for advanced non-small-cell lung cancer (NSCLC) patients was 19.3 months, which did not differ significantly from that of standard chemotherapy [20]. In the present study, the survival of a lung cancer patient treated with erlotinib was 81.2 months, and she is still alive (31 October 2019 at present). The median survival of NSCLC cancer patents was not reached at the end of the observation period. The median survival of pancreatic cancer patients treated with nab-paclitaxel plus gemcitabine or gemcitabine plus S-1 was 8.5 months and 10.1 months, respectively [21,22]. In the present study, the median survival of pancreatic cancer patients was 10.7 months. Despite the fact that most of the present patients were already treated with ≥ second line chemotherapy, median survival in the present study was better than expected. The present results suggest that the ketogenic diet and chemotherapy have a synergistic effect in the treatment of cancer, as seen in the mouse models.

In this study, the KD-ABC score, composed of ALB, BS, and CRP, was proposed, and it stratified the prognosis of cancer patients treated with the ketogenic diet. It is well known that diabetes mellitus is a risk factor [23]. This is the first report to demonstrate that blood glucose levels are related to the prognosis of cancer patients on a ketogenic diet. Only one previous report was found in patients with stage III colorectal cancer receiving adjuvant FOLFOX6 [24]. The ketogenic diet is said to affect glucose metabolism [25] and gene expression [26,27] because of changes in the metabolic environment and ketone bodies acting as signal molecules [28]. The term “ketogenic metabolic therapy” has also been proposed from the ketogenic diet [29,30]. In the present study, blood glucose and insulin levels were significantly decreased, suggesting that the ketogenic diet directly affects the insulin/IGF-1 signaling pathway and PI3K activity. The present data provide various hints for a better understanding of the molecular mechanisms of the ketogenic diet in the treatment of cancer.

The present ketogenic diet regimen had better tolerability, with less nausea, fatigue, and constipation, and it induced stable adherence. Hyperuricemia, hyperlipidemia, and various digestive symptoms were well controlled by drug treatment. In the present study, weight loss was observed, as in another clinical trial [31]. However, fat mass showed significant changes, but not muscle weight using a body composition analyzer. Thus, these results suggest that weight loss is not toxic effect of ketogenic diet because nutrition and muscle mass maintained. It is said that a ketone ratio from 2:1 to 4:1 is needed for children with epilepsy, but it is controversial for adult cancer patients [10]. If our ketogenic diet regimen had also incorporated a higher ketone ratio from 3:1 to 4:1, it might have shown a higher clinical effect on the cancer patients. The present regimen includes a ketone ratio of 2:1 for only one week to induce high ketosis, which affects several metabolic pathways and gene expressions. From the second week, the ketone ratio from 1:1 to 2:1 improved the diet’s tolerability. From three months, the ketone ratio of 1:1 enabled long-term continuity of this study. Our ketogenic diet regimen has also an aspect of mild caloric restriction, which may have influenced the tolerability. Caloric intake of 30 kcal/kg/day and protein intake of 60–70 g/day (1–1.2 g/kg/day) were adopted from the ESPEN guideline for cancer-related malnutrition [2].

The ketogenic diet regimen and the clinical stage of cancer were well controlled, but the present case series study had several limitations. A variety of cancer patients with various treatments depending on the type of cancer were examined without a control group. Despite discussing the research design as to what is a control group compared to a ketogenic diet plus chemotherapy, a randomized, controlled trial is needed to clarify the effect of the ketogenic diet for cancer patients in the near future.

## 5. Conclusions

The present ketogenic diet regimen seems to offer hope as a promising support therapy. Stable adherence and highly reproducible results indicate that this regimen could become the standard diet in chemotherapy trials and for patients with various advanced cancers.

## Figures and Tables

**Figure 1 nutrients-12-01473-f001:**
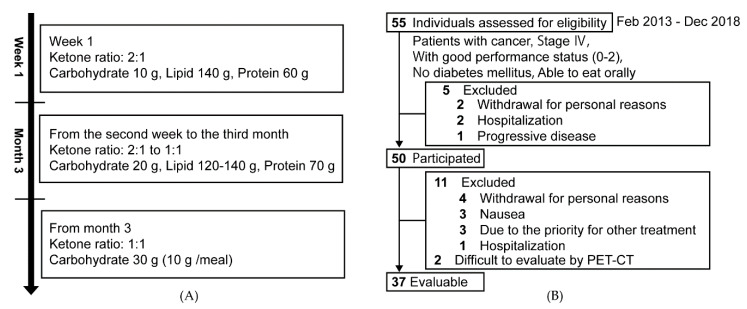
Ketogenic diet for patients with cancer. (**A**) Regimen of the ketogenic diet for cancer patients. The ketone ratio is calculated as: lipid/protein + carbohydrate. Daily caloric intake is 30 kcal/kg ideal body weight. Medium chain triglyceride (MCT) oil and a ketogenic formula are used for calorie supplementation. (**B**) Flow diagram of 37 evaluable patients from among the 55 participants.

**Figure 2 nutrients-12-01473-f002:**
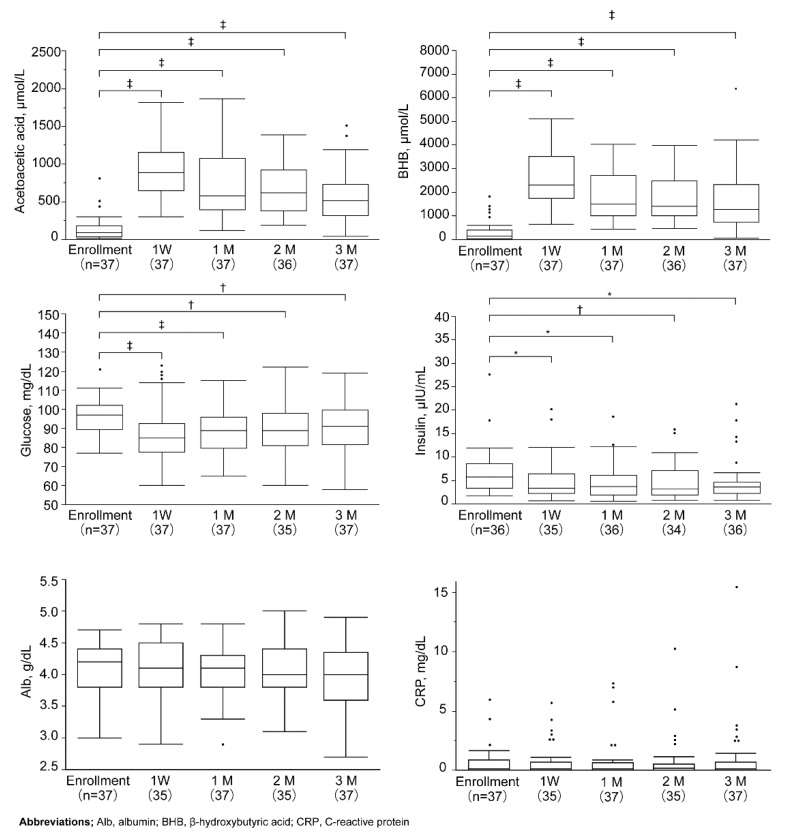
Blood data of patients after the ketogenic diet. Box plots represent medians and interquartile ranges; whiskers indicate range of values that fall within 1.5 box lengths. Footnotes indicate significant differences (Wilcoxon signed-rank test; enrollment vs. 1 week and 1, 2, and 3 months. * *p* < 0.001, † *p* < 0.01, ‡ *p* < 0.05). BHB, β-hydroxybutyric acid; Alb, albumin; CRP, C-reactive protein.

**Figure 3 nutrients-12-01473-f003:**
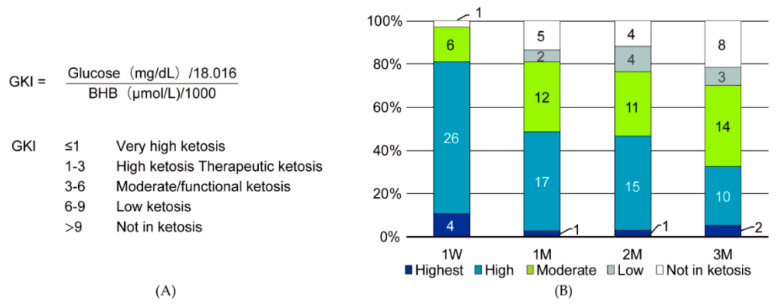
(**A**) Glucose ketone index (GKI) and (**B**) the GKI after the ketogenic diet.

**Figure 4 nutrients-12-01473-f004:**
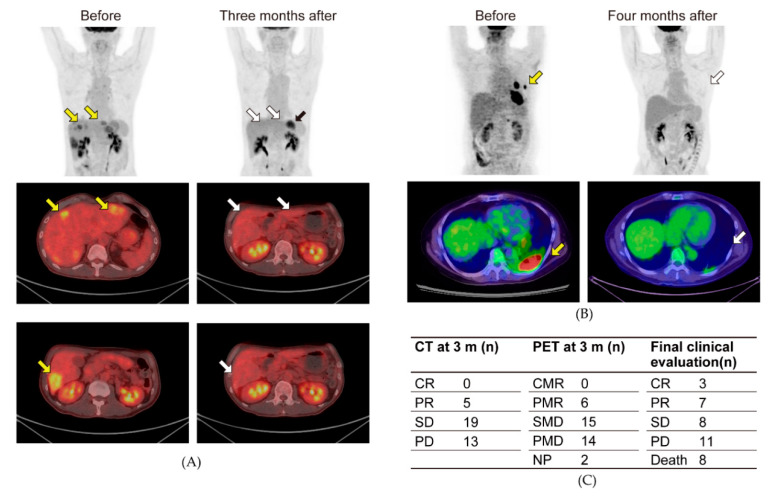
Clinical effect of the ketogenic diet for patients with cancer. Whole body maximal intensity projection images and axial fused PET/CT images before and after the ketogenic diet. (**A**) A 58-year-old man with colorectal cancer. (**B**) A 55-year-old man with lung cancer. Hot spots of 18F-FDG before (yellow arrow) and the same position after the ketogenic diet (white arrow). Physiological hot spot in the stomach (black arrow). (**C**) Response evaluation after the ketogenic diet. Response evaluation with PET/CT at 3 months and clinical evaluation at one year. CT imaging findings were assessed using Response Evaluation Criteria in Solid Tumors (RECIST) version 1.1. CR, complete response; NP, not performed; PD, progressive disease; PR, partial response; SD, stable disease. PET imaging findings were analyzed according to European Organization for Research and Treatment of Cancer (EORTC) criteria. CMR, complete metabolic response; PMR, partial metabolic response; SMD, stable metabolic disease; PMD, progressive metabolic disease.

**Figure 5 nutrients-12-01473-f005:**
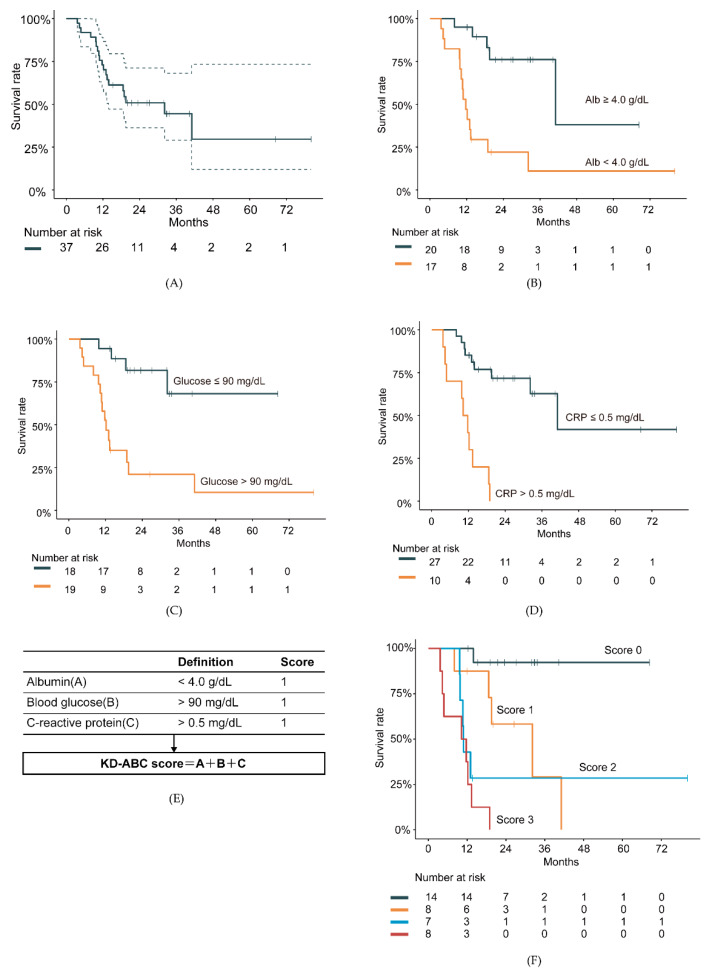
Cumulative survival rates of patients on the ketogenic diet. (**A**) Continuous line shows overall survival of 37 patients. Dotted lines indicate 95% confidence intervals. (**B**) Patients with serum values for albumin (Alb) ≥ or < 4.0 mg/dL (*p* < 0.001); (**C**) glucose (Glu) > or ≤ 90 mg/dL (*p* < 0.001) and (**D**) C-reactive protein (CRP) > or ≤ 0.5 mg/dL (*p* < 0.001). (**E**) Diagram of the ketogenic diet-ABC (KD-ABC) score calculation method. (**F**) KD-ABC scores significantly stratify cumulative survival rates (*p* < 0.001). All *p* values determined by log-rank tests.

**Table 1 nutrients-12-01473-t001:** Background of evaluable patients (*n* = 37).

Age, *y*	54.8 ± 12.6
Sex (male/female), *n*	15/22
Body height, cm	162.5 ± 9.5
Body weight, kg	55.5 ± 13.2
Body mass index, kg/m^2^	20.9 ± 3.7
**Primary cancer**	
Colorectal cancer, *n*	8
Non-small cell lung cancer, *n*	6
Breast cancer, *n*	5
Pancreatic cancer, *n*	4
Head and neck cancer, *n*	3
Bone and soft tissue sarcoma, *n*	3
Ovarian cancer and peritoneal cancer, *n*	2
Endometrial cancer, *n*	1
Bladder cancer, *n*	1
Brain tumor, *n*	1
Biliary tract cancer, *n*	1
Gastric cancer, *n*	1
Prostate cancer, *n*	1
**Treatment history**	
Chemohormonal therapy, *n* (%)	33 (89.2)
Radiation therapy, *n* (%)	13 (35.1)
Surgical therapy, *n* (%)	26 (70.3)

**Table 2 nutrients-12-01473-t002:** Summary of adverse events from enrollment to 3 months.

CTCAE v5.0 Term	Grade				Total *n* = 55 (%)
	G1	G2	G3	G4	
Hyperuricemia	8	24	0	0	32 (58.2)
Hypoglycemia	4	6	0	0	10 (18.2)
Cholesterol high	12	13	1	0	26 (47.3)
Hypertriglyceridemia	11	2	0	0	13 (23.6)
Hyperlipidemia	14	15	0	0	29 (52.7)
Hypokalemia	1	4	0	0	5 (9.1)
Hypocalcemia	0	1	0	0	1 (1.8)
Abdominal pain	0	2	0	0	2 (3.6)
Constipation	0	17	0	0	17 (30.9)
Vomiting	0	1	0	0	1 (1.8)
Nausea	0	1	0	0	1 (1.8)
Stomach pain	0	8	0	0	8 (14.5)
Diarrhea	0	7	0	0	7 (12.7)
Dyspepsia	0	1	0	0	1 (1.8)
Anorexia	0	1	0	0	1 (1.8)
Muscle cramp	0	1	0	0	1 (1.8)
Malaise	0	1	0	0	1 (1.8)
Weight loss	10	2	0	0	12 (21.8)

CTCAE v5.0 Term indicates Common Terminology Criteria for Adverse Events v5.0.

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
