# Peer review of "Promising Effect of a New Ketogenic Diet Regimen in Patients with Advanced Cancer"

_nutrients, 2020, doi:10.3390/nu12051473_

Round 1

Reviewer 1 Report

Evidence from a case study is presented showing therapeutic benefit from a new ketogenic diet regimen in patients with different types of stage IV cancers.  Carbohydrates were restricted to 10 g/day during week 1, 20 g/day from week 2 to 22 three months, and 30 g/day thereafter.  Median survival was 32.2 months, and the 3-year survival rate was 44.5%.  Patient survival was linked to serum levels of Alb, BS, and CRP (ABC).  The authors conclude that their ketogenic diet regimen should be considered a promising support therapy for patients with various advanced cancers.  The data support the conclusions, but a number of issues should be addressed to improve the manuscript.

Major comments

  1. The supplementary materials was not accessible using the www.mdpi.com/xxx/s1 link provided. The authors should also provide a table showing glucose, BHB, and GKI values for each of the 37 patients both before and after the KD treatment.
  2. Some patients were treated with chemotherapy and KD.How many patients were treated with KD alone or did all patients have some combination of treatments? Was diarrhea due to chemotherapy or to the KD?  Which events can be attributed to the KD or other treatments?
  3. An important finding was that the KD reduced fat mass and not muscle mass. This would identify KD weight loss as therapeutic rather than as toxic. Weight loss from chemotherapy or cachexia is toxic weight loss involving muscle wasting.  The authors should elaborate on this further making it clear that the weight loss from their KD treatment is not toxic.
  4. The term “ketogenic metabolic therapy” is now being used as an alternative to ketogenic diet therapy (http://dx.doi.org/10.1016/j.critrevonc.2017.02.016). The authors should also incorporate this term for their study, as the term ‘diet” is not favorably received in medicine.
  5. It is not clear why the authors needed to increase the carbohydrate content gradually. The data clearly shows that patients have the best results in terms of glucose, ketones and  GKI after the first week of diet treatment.  
  6. 26 patients had high cholesterol and 10 patients had hypoglycemia. Were those the same patients? 
  7. Improvement is the English writing could significantly improve the clarity of the manuscript.
  8. The authors should speculate on lines 275-278 if a mild calorie restriction might improve the tolerability and health benefits of their KD, as calorie restriction is often recommended for improving KD therapy.
  9. There are a number of studies missed in the references that should be acknowledged in light of the author’s findings. These include: (https://doi.org/10.1080/02656736.2019.1589584; DOI:10.12691/ajmcr-4-8-8; DOI:10.12691/ajmcr-5-8-3; doi: 10.3389/fnut.2020.00021; )

Minor comments

  1. A list of abbreviations is needed Alb, CRP, mFOLFOX6 and Pmab, NSCLC, etc.
  2. On line 81, “baseline” weight should be replace “real” weight.
  3. More detail is needed for The MCT oil and ketone formula on lines 89-90.
  4. The information presented on lines 93-94 require more detail.
  5. The sentences on lines 121-125 should be rewritten as a method, not a directive.
  6. More information is needed on the 11 patients that dropped out (line 131).

Author Response

Major comments

  1. The supplementary materials was not accessible using the www.mdpi.com/xxx/s1 link provided. The authors should also provide a table showing glucose, BHB, and GKI values for each of the 37 patients both before and after the KD treatment.

Thank you for your important suggestion. We added the supplemental Table (Table S4) showing glucose, BHB, and GKI values for each of 37 patients before and after KD treatment. We added the sentence on lines 150 -151.

   We showed glucose, BHB, and GKI values for each of 37 patients in detail

 (Table S4).

  1. Some patients were treated with chemotherapy and KD. How many patients were treated with KD alone or did all patients have some combination of treatments? Was diarrhea due to chemotherapy or to the KD? Which events can be attributed to the KD or other treatments?

Thank for your important question. We already described lines 136. 33 patients received chemo-hormonal therapy (89%) at the enrollment of KD. Our patients showed the performance status <2. Thus, our patients were not in a palliative care state. Basically, our patients received some combination therapy with KD, not KD alone.

KD mainly induced constipation. In fact, GSRS score of constipation increased after KD in our study. However, patients sometimes showed diarrhea due to amounts of MCT oils. In that case we adjusted the amounts of MCT oils. Diarrhea induced by chemotherapy occurs several days after chemotherapy. It is easy to distinguish the cause KD or chemotherapy.

  1. An important finding was that the KD reduced fat mass and not muscle mass. This would identify KD weight loss as therapeutic rather than as toxic. Weight loss from chemotherapy or cachexia is toxic weight loss involving muscle wasting. The authors should elaborate on this further making it clear that the weight loss from their KD treatment is not toxic.

Thank you for your important suggestion. We added the sentences on lines 287-288.

   Thus, these results suggest that weight loss is not toxic effect of ketogenic diet because nutrition and muscle mass maintained.

  1. The term “ketogenic metabolic therapy” is now being used as an alternative to ketogenic diet therapy (http://dx.doi.org/10.1016/j.critrevonc.2017.02.016). The authors should also incorporate this term for their study, as the term ‘diet” is not favorably received in medicine.

Thank you for your nice suggestion about a new concept. It is interesting words. However, I’m sorry that it is difficult to change every words “ketogenic diet” to “ ketogenic metabolic therapy”. We added the sentences on lines 278-279 and the reference 29. 

   The term "ketogenic metabolic therapy" has also been proposed from the

ketogenic diet [29,30].

  1. Winter SF, Loebel F, Dietrich J. Role of ketogenic metabolic therapy in

malignant glioma: A  systematic review. Crit Rev Oncol Hematol.

2017 ;112:41-58. doi: 10.1016/j.critrevonc.2017.02.016.

  1. It is not clear why the authors needed to increase the carbohydrate content gradually. The data clearly shows that patients have the best results in terms of glucose, ketones and GKI after the first week of diet treatment.

We considered the safety and tolerability of KD for adult cancer patients. Restrict of carbohydrate is needed for the clinical effect on the cancer patients. But, carbohydrate restriction to 10 g/ day shows that the variety of the meal menu is extremely reduced. This is directly related to the tolerability of adult cancer patients. On the other hands, carbohydrate restriction to 20 g/ day increases the variety of the meal menu. We created a new ketogenic diet regime with a focus on continuity.

  1. 26 patients had high cholesterol and 10 patients had hypoglycemia. Were those the same patients?

We counted 26 patients with high cholesterol in detail.

19 patients showed high cholesterol only.

7 patients showed high cholesterol and hypoglycemia.

3 patients showed hypoglycemia only.

  1. Improvement is the English writing could significantly improve the clarity of the manuscript.

Thank you for your suggestion. Our draft was checked by native English speaker repeatedly.

  1. The authors should speculate on lines 275-278 if a mild calorie restriction might improve the tolerability and health benefits of their KD, as calorie restriction is often recommended for improving KD therapy.

In general, Japanese patients are smaller and thinner than western patients. Our new ketogenic diet regimen showed a suitable for Japanese patients. From the points of view of western people, our new regimen might be a mild calorie restriction.

We added the sentences on lines 295-296.

 Our ketogenic diet regimen has also an aspect of mild caloric restriction, which may have influenced the tolerability.

  1. There are a number of studies missed in the references that should be acknowledged in light of the author’s findings. These include: (https://doi.org/10.1080/02656736.2019.1589584; DOI:10.12691/ajmcr-4-8-8; DOI:10.12691/ajmcr-5-8-3; doi: 10.3389/fnut.2020.00021; )

Thank you for your nice suggestion. We added the sentences on lines 278-279 and the reference 30.

   The term "ketogenic metabolic therapy" has also been proposed from the

ketogenic diet [29,30].

  1. Seyfried TN, Mukherjee P, Iyikesici MS, Slocum A, Kalamian M, Spinosa JP,

et al. Consideration of Ketogenic Metabolic Therapy as a Complementary or

Alternative Approach for Managing Breast Cancer.Front Nutr. 2020;11;7:21. doi:

10.3389/fnut.2020.00021.

Minor comments

  1. A list of abbreviations is needed Alb, CRP, mFOLFOX6 and Pmab, NSCLC, etc.

Abbreviation lists showed in each tables and figures.

In the draft we added on lines 262 (NSCLC).

  1. On line 81, “baseline” weight should be replace “real” weight.

We changed baseline to real on lines 82.

  1. More detail is needed for The MCT oil and ketone formula on lines 89-90.

We added the detail of MCT oil and ketone formula on Table S1 and added the sentences on lines .89-92.

Medium chain triglyceride (MCT) oil about 50-80 g/day and a ketogenic formula about 30 g /day were used. The contents of a ketogenic formula were shown in Table S1.

  1. The information presented on lines 93-94 require more detail.

We added the sentences on lines 94-96.

Registered dieticians provided nutritional guidance according to general condition and laboratory data at enrollment, 7 days after enrollment, Month 1, 2, and 3. After that, patients received nutritional guidance every 1-2 months.

  1. The sentences on lines 121-125 should be rewritten as a method, not a directive.

I’m sorry this is a simple mistake. We deleted lines 121-125.

  1. More information is needed on the 11 patients that dropped out (line 131).

We showed the detail of 11 patients in figure 1B. We added words on lines 131.

Reviewer 2 Report

See my comments throughout the text.

Author Response

  1. Many oncology-related protocols for the ketogenic diet use a much stricter ratio (e.g.: 4:1 or 3:1) and a calorie restriction. Therefore, patients might have found some benefits, but likely the result could have been more relevant if ratio and calorie restriction were higher.

Thank you for your important suggestion. We added the sentences on lines 288 - 291.

  but it is controversial for adult cancer patients [10]. If our ketogenic diet regimen had also incorporated a higher ketone ratio from 3:1 to 4:1, it might showed a higher clinical effect on the cancer patients.

  1. Authors should briefly specify the reason why these patients dropped.

Thank you for your suggestion. We showed the detail of 11 patients in figure 1B. We added words on lines 131.

  1. Many studies have been published regarding this feature. I suggest the authors to elaborate this concept researching in literature. In addition, there have been several studies on prolonged 3:1 and 4:1 ratio ketogenic diets for cancer patients. Authors should implement also this feature.

 Thank you for your important suggestion. We added the sentences on lines 288 - 291.

  but it is controversial for adult cancer patients [10]. If our ketogenic diet regimen had also incorporated a higher ketone ratio from 3:1 to 4:1, it might showed a higher clinical effect on the cancer patients.

  1. Again, this feature is controversial. I invite the authors to collect more pieces of information in the literature.

 Thank you for your important suggestion. We added the sentences on lines 288 - 290.

  but it is controversial for adult cancer patients [10]. If our ketogenic diet regimen had also incorporated a higher ketone ratio from 3:1 to 4:1, it might showed a higher clinical effect on the cancer patients.
